# DOES TRAINING WITH SYNTHETIC DATA TRULY PROTECT PRIVACY?

**Yunpeng Zhao**
National University of Singapore
yunpeng.zhao@u.nus.edu

**Jie Zhang**
ETH Zurich
jie.zhang@inf.ethz.ch

## ABSTRACT

As synthetic data becomes increasingly popular in machine learning tasks, numerous methods—without formal differential privacy guarantees—use synthetic data for training. These methods often claim, either explicitly or implicitly, to protect the privacy of the original training data. In this work, we explore four different training paradigms: coreset selection, dataset distillation, data-free knowledge distillation, and synthetic data generated from diffusion models. While all these methods utilize synthetic data for training, they lead to vastly different conclusions regarding privacy preservation. We caution that empirical approaches to preserving data privacy require careful and rigorous evaluation; otherwise, they risk providing a false sense of privacy.

## 1 INTRODUCTION

Synthetic data is increasingly utilized for training machine learning (ML) models, especially in situations where real-world data is scarce, sensitive, costly to obtain, or subject to regulations such as GDPR (GDPR.eu). Synthetic data is particularly beneficial in scenarios where data distributions are atypical, such as in federated learning with non-IID data (Zhang et al., 2023c), long-tailed learning (Shin et al., 2023), and continual learning (Meng et al., 2024). It enables the creation of diverse datasets that include edge cases or rare events that may be underrepresented in real-world data. Consequently, training models with synthetic data has proven beneficial for enhancing model robustness and adaptability across a wide range of real-world scenarios.

Many empirical methods—*without* formal differential privacy guarantees—rely on synthetic data for training, such as coreset selection (Feldman, 2020), dataset distillation (Wang et al., 2018), data-free knowledge distillation (Yin et al., 2020), and synthetic data generated from diffusion models (Yuan et al., 2024). These approaches involve training ML models using proxy data[1] instead of the original private training data. This proxy data can be directly sampled from private sources (Guo et al., 2022; Mirzasoleiman et al., 2020) or out-of-distribution sources (Wang et al., 2023), iteratively optimized (Zhang et al., 2023d; Zhao et al., 2020), or generated using GANs (Karras et al., 2019) and diffusion models (Rombach et al., 2022). Since the model may never encounter any private training data and the synthetic images are often visually distinct from the original private data, these methods often claim to *preserve privacy* while still maintaining satisfactory performance.

In this work, we aim to address the following question:

*Does training with synthetic data truly protect privacy?*[2]

To rigorously measure the privacy leakage of empirical methods trained on synthetic data, we use membership inference attacks (Shokri et al., 2017) as a privacy auditing tool. We provide a systematic privacy evaluation on these four training paradigms. For each training paradigm, we interact only with the final model trained on synthetic data, and then determine whether a particular data point was part of the *private* training dataset.

We also provide a fair comparison with theoretical defenses with differential privacy, such as DPSGD (Abadi et al., 2016), and always report the privacy leakage of these methods in the *worst*

---

[1]For simplicity, for the rest of the paper, we will always use the term "synthetic data" (also for coreset).

[2]Existing works on generating synthetic data with differential privacy are outside the scope of this work.

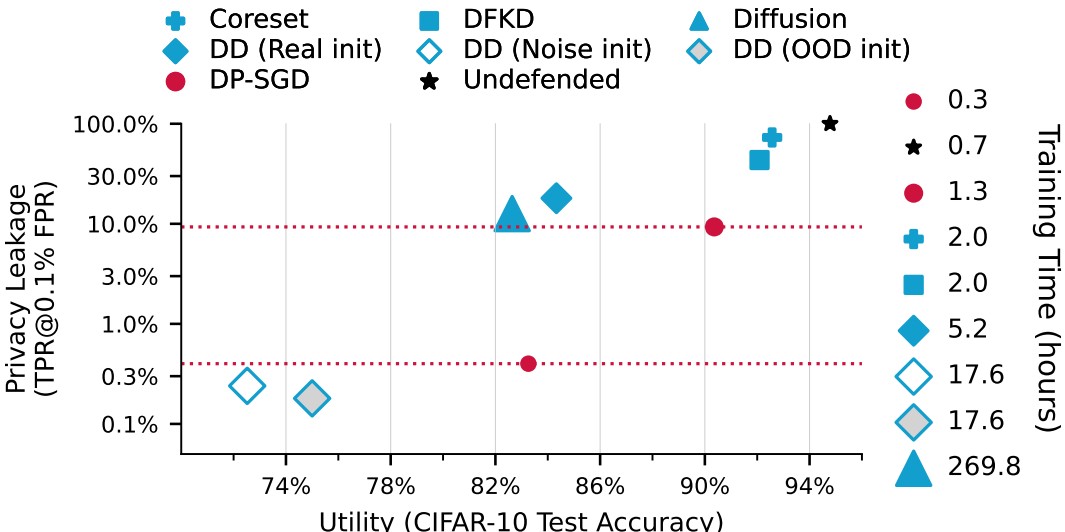

Figure 1: A rigorous evaluation of privacy leakage in models trained with synthetic data. We compare the privacy-utility tradeoff and efficiency of four training paradigms—coreset selection, dataset distillation (DD), data-free knowledge distillation (DFKD), and synthetic data generated from diffusion models—against DPSGD.

*case.* An ideal defense should strike a good balance between privacy and model utility, while also being efficient.

As shown in Figure 1, we employed the most rigorous evaluation framework described in Aerni et al. (2024), training multiple shadow models to accurately evaluate membership inference success, specifically focusing on the true positive rate (TPR) at very low false positive rates (FPR) on the CIFAR-10 dataset. We ensure that each method achieves at least approximately 73% accuracy on the test set, as models with low accuracy may lead to misleading conclusions (e.g., overfitting) and are not meaningful.

Similar to the findings in Aerni et al. (2024), almost all empirical training paradigms do not provide stronger privacy protection than the basic differential privacy baseline—DPSGD. Another interesting finding is that for some dataset distillation methods, the success rate of membership inference attacks is very low, yet the synthetic data *visually* resembles the private data almost entirely (see Figure 7). We argue that this does not constitute meaningful privacy protection. Based on these findings, we caution that empirical approaches to preserving data privacy require careful and rigorous evaluation; otherwise, they risk providing misleading conclusions.

## 2 RELATED WORK

**Membership Inference Attack** Membership inference attack (Shokri et al., 2017) is a canonical approach to estimating privacy leakage, which aims to determine whether a given example was part of the training set. State-of-the-art attack methods frame membership inference as a hypothesis testing problem and evaluate privacy leakage by reporting the true positive rate at very low false positive rates (Carlini et al., 2022a). Recent work (Aerni et al., 2024) suggests that many empirical defenses fail to evaluate privacy leakage in worst-case scenarios; instead, they often report average-case privacy leakage, which can significantly underestimate actual privacy risks. Since privacy is not an average metric, we adopt this approach to provide a rigorous and careful evaluation of privacy.

**Training on Synthetic Data** Many studies have highlighted that training models on synthetic data can be remarkably effective in scenarios where data collection or distribution is challenging. For instance, in long-tailed learning and federated learning, generative models can be used to augment data, resulting in a more balanced distribution (Shin et al., 2023; Zhang et al., 2023c). Similarly, in continual learning, synthetic data is adopted to help mitigate the problem of catastrophic forget-

ting (Meng et al., 2024; Zhang et al., 2023b; Shin et al., 2017). In scenarios where only the target model is available without access to its private training data, generative models can be used to create a synthetic dataset to train a substitute model, which can achieve similar performance to the target model (Zhang et al., 2023a; Lopes et al., 2017). Furthermore, Yuan et al. (2024) demonstrated that a model trained solely on synthetic data can perform well on the ImageNet test set. Cazenavette et al. (2022); Guo et al. (2024) showed that even when the original training set is condensed to just 1% of its size as synthetic data, it is still possible to train a well-performing model on CIFAR10 test set. These examples illustrate the effectiveness of synthetic data, but in this work, we are more curious about whether these models trained with synthetic data actually protect privacy.

A related study Aerni et al. (2024) rigorously evaluates several empirical membership inference defenses, including techniques such as label smoothing Chen & Pattabiraman (2024), training loss calibration Chen et al. (2022), and self-supervised learning. While our experiments build on their evaluation framework, we extend this work by specifically investigating empirical methods that train ML models using *synthetic data*.

## 3 MISLEADING PRIVACY EVALUATIONS ON SYNTHETIC DATA

### 3.1 EMPIRICAL DEFENSES: TRAINING MODELS WITH SYNTHETIC DATA

In this section, we provide a brief introduction to the four training paradigms for generating synthetic data that we studied. A comparison of these training paradigms is presented in Table 1, and we show our evaluation setup in Figure 2.

Table 1: Comparison of methods with respect to privacy considerations for the final model. "Private Data" indicates that the final model is trained directly on some private data, and "partial" means that some method requires it.

| Training Paradigm | Private Data | Public Data | Generative Model | Teacher Model |
|---|---|---|---|---|
| Coreset Selection[1] | ✓ | X | X | X |
| Dataset Distillation[2] | X | partial | X | partial |
| DFKD[3] | X | X | ✓ | ✓ |
| Diffusion[4] | X | ✓ | ✓ | X |

[1](Toneva et al., 2019)  [2](Guo et al., 2024; Cazenavette et al., 2022; Zhao & Bilen, 2021; 2023)
[3](Fang et al., 2022)  [4](Yuan et al., 2024)

**Coreset Selection**   Coreset selection (Toneva et al., 2019; Welling, 2009) aims to extract a compact and informative subset $D_{\text{core}}$ that captures the essential characteristics of the original private training set $D_{\text{train}}$. The selection algorithm begins by evaluating the significance of each data instance in $D_{\text{train}}$, scoring them based on criteria such as the frequency of forgetting events during training (Toneva et al., 2019) or their distance from the cluster center in feature space (Welling, 2009). After scoring, $D_{\text{core}}$ is formed by selecting the top $k$ samples with the highest scores, which can be expressed as:

$$D_{\text{core}} = \{x_i \in D_{\text{train}} \mid \text{score}(x_i) \in \text{Top}(\text{score}(D_{\text{train}}), k)\},$$

where $\text{score}(\cdot)$ is the scoring metric and $\text{Top}(\cdot, k)$ selects the top $k$ highest scores. The selected coreset $D_{\text{core}}$ then replaces the original private dataset for model training. It is obvious that the samples in the subset still face the potential risk of privacy leakage.

**Dataset Distillation (DD)**   DD (Wang et al., 2018; Zhao & Bilen, 2023; 2021; Cazenavette et al., 2022; Guo et al., 2024) aims to *learn* a small set of informative synthetic images $D_{\text{syn}}$ from a large training dataset $D_{\text{train}}$, such that a neural network trained on $D_{\text{syn}}$ achieves similar or comparable generalization performance to a network trained on the original dataset. The process begins by initializing images in $D_{\text{syn}}$ from real images or random noise. After initialization, the goal of DD is formulated as an optimization problem, seeking to minimize the discrepancy between the synthetic and original datasets by aligning their effects on the neural network $\phi_\theta$ parameterized by $\theta$. The optimization objective $\mathcal{L}_{\text{dd}}$ may involve differences between gradients of two sets of network

Figure 2: We evaluate the privacy leakage of private training data in the worst-case scenario for each training paradigm, only interacting with the final model trained on synthetic data.

parameters (Zhao & Bilen, 2021), disparities in feature distributions across multiple sampled embedding spaces (Zhao & Bilen, 2023), or variations in model training trajectories (Cazenavette et al., 2022; Guo et al., 2024):

$$\min_{D_{\text{syn}}} \mathcal{L}_{\text{dd}}(\phi_\theta(D_{\text{syn}}), \phi_\theta(D_{\text{train}})).$$

After convergence, the synthesized dataset $D_{\text{syn}}$ can be used to train models. However, potential privacy risks may arise during the initialization stage if real images from the private dataset $D_{\text{train}}$ are used to initialize $D_{\text{syn}}$. A critical concern is whether the subsequent DD optimization process adequately safeguards the privacy of these initial samples.

**Data-Free Knowledge Distillation (DFKD)**  DFKD (Yin et al., 2020; Fang et al., 2022) transfers knowledge from a teacher model $f_{\text{teacher}}$ to a student model $f_{\text{student}}$ using synthetic data that approximates the private training data. First, $f_{\text{teacher}}$ is trained on $D_{\text{train}}$. Then, a generative model $G$ creates synthetic data $\{x_i\}_{i=1}^N$ to match the teacher model's statistical properties, minimizing an inversion loss:

$$\min_{G} \mathcal{L}_{\text{gen}}(G, f_{\text{teacher}}, D_{\text{train}}).$$

The student model is trained on this synthetic data and the teacher's predictions, minimizing the distillation loss:

$$\min_{f_{\text{student}}} \mathcal{L}_{\text{distill}}(f_{\text{student}}, \{x_i\}_{i=1}^N, f_{\text{teacher}}).$$

This process iterates until convergence. Privacy risks may arise if the teacher model memorizes specific instances from $D_{\text{train}}$.

**Synthetic Data from Fine-Tuned Diffusion Models**  This approach (Yuan et al., 2024) involves fine-tuning a stable diffusion model on a private dataset to learn its distribution. Leveraging the model's powerful generative capabilities, synthetic data is then generated and used to train models. Potential privacy leakage may occur if the fine-tuned diffusion model retains or reveals sensitive information from the private dataset, especially if the generated synthetic data closely resembles the original data.

### 3.2 PRIVACY LEAKAGE ON MOST VULNERABLE SAMPLES

Our motivation is to properly evaluate privacy leakage in ML models trained on synthetic data. First, we will highlight some misleading evaluations that give a false sense of privacy. Then, we will demonstrate that models tend to strongly memorize the most vulnerable samples, which is why training with synthetic data can lead to privacy leakage.

**Misleading Privacy Evaluations on Synthetic data**  Since privacy is not a metric that can be averaged (Steinke & Ullman, 2020), Aerni et al. (2024) advocates for rigorous evaluation by reporting privacy leakage on the most vulnerable samples in a dataset, rather than the average case. To enhance computational efficiency, they suggest using *mislabeled data*, which approximates the privacy leakage of the most vulnerable samples. However, none of the existing work has systematically evaluated privacy leakage in the worst-case scenario for synthetic data.

Previous work (Dong et al., 2022) claims that dataset distillation can significantly improve data privacy, but the level of privacy protection is not evaluated in a correct way (Carlini et al., 2022b).

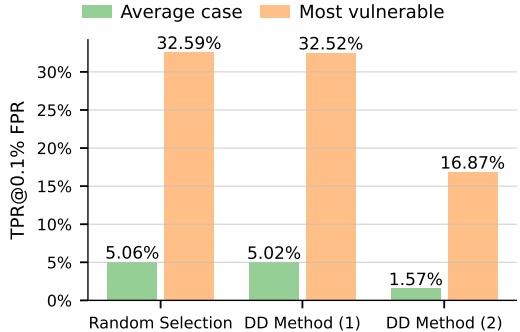

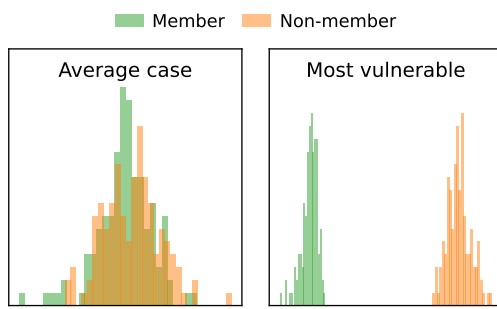

Figure 3: Failing to report privacy leakage on the most vulnerable data provides a *false* sense of privacy. We investigate three different defenses: one based on coreset selection and two based on dataset distillation.

Figure 4: ML models tend to strongly memorize the most vulnerable data. We demonstrate this by presenting the loss distribution for both members and non-members, comparing average-case data with worst-case data.

Some studies (Hao et al., 2021; Dong et al., 2022) claim that training on synthetic data protects privacy because synthetic data is visually dissimilar to the private data. However, all these methods give a false sense of privacy, which we further demonstrate in Section 4.

We begin with a toy experiment on coreset selection and dataset distillation[3]. In Figure 3, we demonstrate that for coreset selection and dataset distillation, failing to report privacy leakage on the most vulnerable samples can severely underestimate the true privacy risk. Thus, in this work, we always report privacy leakage in the worst case[4] to avoid underestimating the privacy risk.

**Strong Memorization on Vulnerable Data**  ML models can memorize sensitive information from their training data, particularly for vulnerable samples such as mislabeled data or outliers (Feldman & Zhang, 2020). In Figure 4, we show that for a mislabeled data point, models can strongly memorize it, leading to a significant difference in the loss distribution for this sample when it is in the training set versus when it is not[3]. This discrepancy makes MIA much easier. Therefore, later we will show that no matter how carefully synthetic data training is designed, the strong memorization capabilities of ML models still make it easy to perform MIA on the most vulnerable samples.

## 4 CASE STUDIES: PRIVACY LEAKAGE IN TRAINING WITH SYNTHETIC DATA

In this section, we conduct a systematic and rigorous evaluation of privacy leakage across four training paradigms based on synthetic data. We start by introducing the experimental setups. Subsequently, we will discuss the potential privacy leakage inherent to each method. Finally, we will compare all these methods with a differential privacy baseline (i.e., DPSGD) with respect to model utility, privacy preservation, and computational efficiency.

### 4.1 EVALUATION SETUP

**Evaluation Metric**  Empirical methods designed to preserve data privacy require careful scrutiny. To ensure precise measurement of privacy leakage in models trained with synthetic data, we employ the state-of-the-art membership inference attack LiRA (Carlini et al., 2022a)) and follow the setup from (Aerni et al., 2024). In this setup, we report the privacy leakage by evaluating the true positive rate at a low false positive rate on the most vulnerable samples (e.g., mislabeled data).

**Experimental Setup**  We conduct all experiments on CIFAR-10 (Krizhevsky & Hinton, 2009), as all training methods are scalable to CIFAR-10 and achieve good test accuracy. We designate

---

[3]See Appendix C for detailed experimental setup.

[4]By default, we use **mislabeled data** as strong canaries to simulate the most vulnerable samples, but even stronger canaries may exist, offering a better capture of privacy leakage.

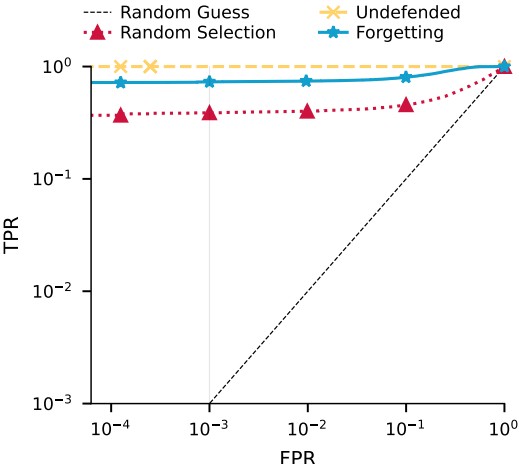
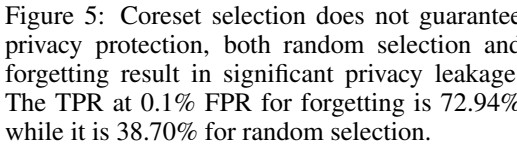

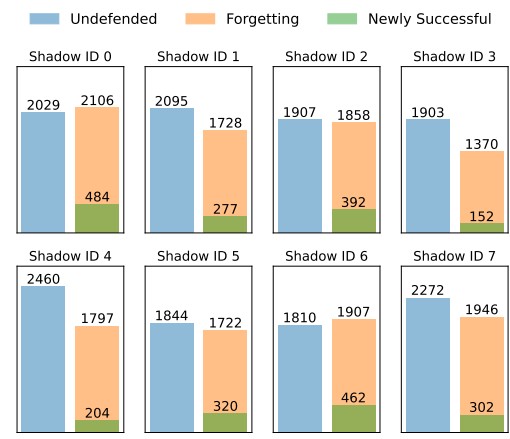

Figure 5: Coreset selection does not guarantee privacy protection, both random selection and forgetting result in significant privacy leakage. The TPR at 0.1% FPR for forgetting is 72.94% while it is 38.70% for random selection.

Figure 6: MI success rate for both the full dataset and the coreset. "Newly successful" denotes samples that are typically very safe when trained on the full dataset but experience a significant increase in privacy leakage once included in the coreset.

500 random data points as "audit samples" on which we evaluate membership inference, and we use mislabeled data as strong canaries to simulate worst case data; the remaining 49,500 samples are always included in every model's training data. For each method, we train 32 shadow models, ensuring that each audit sample is included in the training data of 16 models. For all defenses, we consistently adopt ResNet-18 (He et al., 2016) as the network architecture of shadow models. We report the performance of these methods across three dimensions: privacy leakage (TPR@0.1% FPR), model utility (test accuracy), and efficiency (training time). A strong defense should achieve a balanced tradeoff among privacy, utility, and efficiency. The implementation details of all discussed defenses are provided in Appendix A.

## 4.2 Privacy Leakage in Coreset Selection

We begin by evaluating coreset selection methods to explore whether privacy protection can be achieved by selecting a representative subset to replace the entire private dataset for model training. We compare two representative methods: random selection and forgetting (Toneva et al., 2019), against the undefended baseline. The coreset size is consistently set to 20,000. The average test accuracy across 32 shadow models is 94.78% for the undefended baseline, 92.57% for forgetting, and 90.56% for random selection.

In Figure 5, we find that neither random selection nor forgetting is able to protect the membership privacy of auditing samples as the private subset is directly used for training.

One interesting finding is that the forgetting defense achieves better utility than random selection by selecting informative samples that are easily forgotten during training. However, its privacy leakage is significantly higher. This is likely due to the forgetting method's tendency to select mislabeled samples. While random selection includes roughly 40% of mislabeled data, the forgetting method increases this to 74%, suggesting that these vulnerable data points also contribute to improved generalization (Feldman & Zhang, 2020), at the cost of reduced privacy.

Another interesting finding is that even in average case evaluations, some samples that would typically be very safe (when trained on the entire dataset) experience a significant increase in privacy leakage once selected for the coreset. This suggests that protecting the privacy of non-coreset samples can degrade the privacy of other samples. In worst-case evaluation scenarios, this becomes even more concerning, as no sample would be willing to sacrifice its privacy for the benefit of others. We illustrate this in Figure 6, where users whose privacy appears least at risk may be the most likely not to request their data be included in a coreset.

Table 2: A rigorous evaluation of privacy leakage on four dataset distillation methods. '—' indicates that the model's utility is significantly lower and thus less practical to use.

| Methods | Initialization | Test Accuracy | TPR@0.1% FPR |
|---|---|---|---|
| DM[1] | Private | 84.33% | **18.02**% |
| | Noise | 72.52% | 0.24% |
| | OOD | 75.00% | 0.18% |
| DSA[2] | Private | 84.43% | **17.74**% |
| | Noise | — | — |
| | OOD | — | — |
| MTT[3] | Private | 81.81% | 1.51% |
| | Noise | — | — |
| | OOD | 80.42% | 0.18% |
| DATM[3] | Private | 84.54% | 1.29% |
| | Noise | — | — |
| | OOD | — | — |

[1] Distribution Matching [2] Gradient Matching [3] Trajectory Matching

## 4.3 PRIVACY LEAKAGE IN DATASET DISTILLATION

Current work evaluating the privacy leakage of dataset distillation often contains some flaws. These methods face issues in empirical evaluation, such as using improper averaging metrics (e.g., AUC instead of TPR) (Chen et al., 2023), evaluating models with low performance (around 60% test accuracy), or applying incorrect theoretical analyses (Dong et al., 2022). Furthermore, none of these methods are evaluated under worst-case scenarios. Such non-rigorous evaluations can result in misleading conclusions (Carlini et al., 2022b).

In this work, we conduct experiments on four representative DD methods to rigorously evaluate their privacy leakage, including DM (Zhao & Bilen, 2023), DSA (Zhao & Bilen, 2021), MTT (Cazenavette et al., 2022), and DATM (Guo et al., 2024). We tune the model performance to around 80% test accuracy and evaluate these methods under worst-case scenarios to capture the full extent of potential privacy leakage.

Popular methods in dataset distillation aim to match the behavior of models trained on synthetic data with those trained on private data. This matching can be based on distribution, gradients, or training trajectories. Additionally, synthetic data can be initialized using real private data, out-of-distribution (OOD) data, or random noise. These factors can lead to significantly different levels of privacy leakage. We present the overall results in Table 2 and provide a detailed analysis of how each factor impacts privacy leakage in the following discussion.

**The initialization of synthetic data affects privacy** As shown in Table 2, initializing synthetic data with private data leads to higher privacy leakage in DM compared to initialization with OOD data or random noise. Consequently, while initializing with OOD data or random noise offers better privacy protection, it comes at the cost of substantial reductions in model performance.

To further illustrate why initialization with private data results in greater privacy leakage, we analyze the privacy risks associated with canaries used during the synthetic data initialization process. As shown in Table 3, the mem-

Table 3: The MI success rate on initialization and non-initialization canaries.

| Method | Success rate (%) | |
|---|---|---|
| | Init | Non-init |
| DM | **88.89** | 0.16 |
| DSA | **88.70** | 0.09 |
| MTT | 7.02 | 0.11 |
| DATM | 36.05 | 0.24 |

bership inference success rate for canaries involved in initialization is considerably higher than for those not used during initialization. For canaries present in the training set but excluded from initialization, the success rate is nearly indistinguishable from random guessing. This finding empirically demonstrates that privacy risks are primarily introduced during the initialization phase, as the model tends to strongly memorize these initialization canaries.

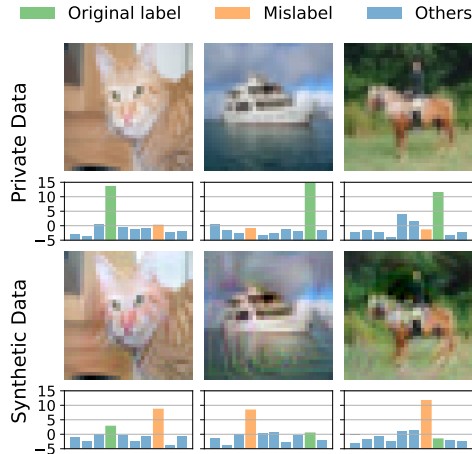
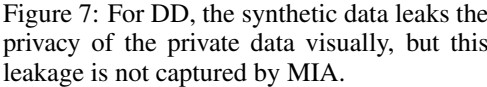

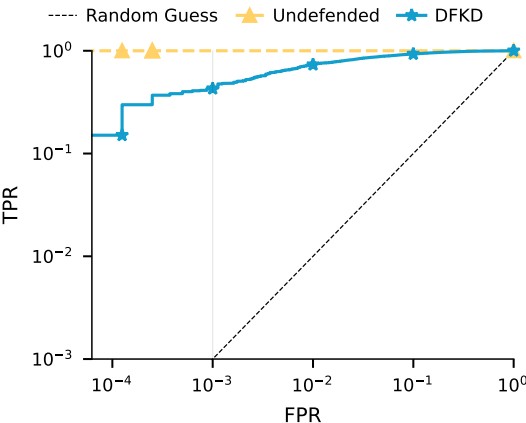

Figure 7: For DD, the synthetic data leaks the privacy of the private data visually, but this leakage is not captured by MIA.

Figure 8: For DFKD, although the student model is never trained on private data and is only trained on synthetic data, due to knowledge distillation, it still leaks a significant amount of privacy.

**Visual privacy leakage in trajectory matching with private initialization**   As demonstrated in Table 2, trajectory-based methods seem to provide superior privacy protection compared to DM and DSA, even when private initialization is utilized. This can be attributed to the fact that trajectory-based approaches, such as MTT and DATM, initially train a teacher model on the original private data and save the training trajectories. During each optimization epoch, a network is selected from the teacher trajectory and trained for several steps using synthetic data, with the objective of matching the trained model's parameters to the teacher trajectory. Notably, MTT and DATM focus on aligning only the early to mid-training trajectories—stages where the model has not yet memorized canaries.

However, **a low MI success rate could also provide a false sense of privacy.** In Figure 7, we visualize both the synthetic data and the private data, which clearly demonstrates visual privacy leakage that is not captured by MIA. We present the logits output by the model trained on synthetic data, for both the private data (canaries) used for initialization and the synthetic data itself. This indicates that while the model can strongly memorize the synthetic data, it generalizes well on the canaries, which results in a low MI success rate. A potential explanation for this phenomenon is that trajectory matching effectively introduces a "weird" data augmentation on the private data, which is hard to estimate. This aspect is not captured during MIA, which typically employs common augmentation techniques like random flip and random crop.

In summary, despite the low MI success rate, MTT entirely compromises the privacy of private data through visual leakage. A potential defense is to use MTT initialized with OOD data, which achieves a low MI success rate (close to random guessing) while offering stronger visual privacy protection (see Figure 13). However, this approach results in reduced model performance, with test accuracy dropping to 80.42%.

## 4.4 PRIVACY LEAKAGE IN DATA-FREE KNOWLEDGE DISTILLATION

Similar to previous work (Aerni et al., 2024), in Figure 8, we also find that since the teacher model can strongly memorize canaries, during the distillation stage, the student model can also somewhat memorize canaries, leading to significant privacy leakage.

We further visualize the synthetic data and private data in Figure 9, revealing an interesting finding: even when a synthetic image appears completely different from the original private image, it can still somehow trigger the teacher model's memorization of the canary, thereby leaking privacy. The SSIM (Wang et al., 2004) score is only 0.1661, yet the logits distribution shows an extremely high similarity, with a correlation of nearly 0.999.

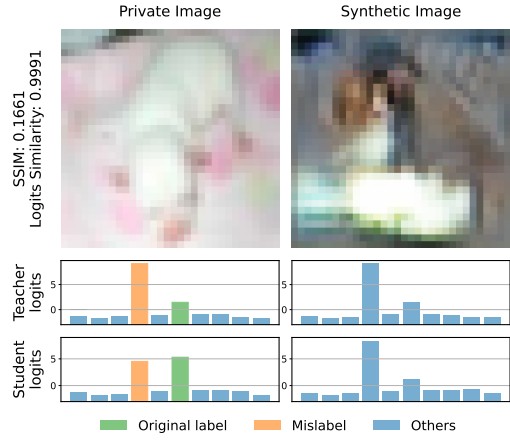 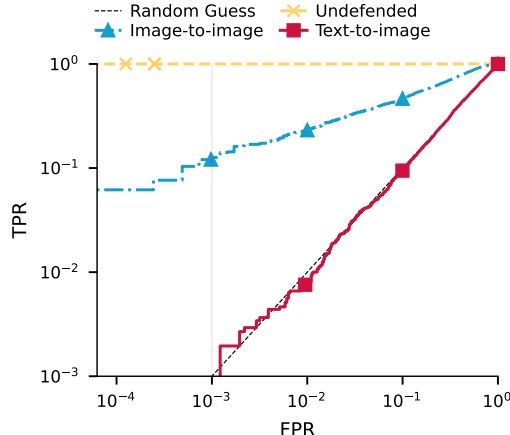

Figure 9: For DFKD, synthetic data can activate the teacher model's memorization of private data, even if the synthetic data appears visually distinct, leaking privacy to student.

Figure 10: For synthetic data from a fine-tuned diffusion model, image-to-image generation still leaks privacy, while text-to-image results in predictions close to random guessing.

In detail, the generator in DFKD is capable of producing synthetic images that, while visually distinct, bear a strong resemblance in logits to the mislabeled data. These synthetic images activate the memories of corresponding canaries encoded in the teacher model's parameters. The distillation process attempts to align the student model's logits distribution with that of the teacher for the same images. Consequently, the student model may learn some private knowledge from the teacher, including some aspects of the teacher's memorization of mislabeled data.

In summary, the visual dissimilarity between synthetic data and private data does not necessarily guarantee privacy protection.

## 4.5 PRIVACY LEAKAGE IN SYNTHETIC DATA FROM DIFFUSION MODELS

A representative method for this training paradigm is presented by (Yuan et al., 2024), which involves three stages: (1) first, fine-tuning a stable diffusion model on a private dataset; (2) then using the fine-tuned model for image-to-image generation by passing a text prompt and an initial image from the private data to condition the generation of new images; (3) finally, replacing all private data with the generated synthetic images (while keeping the original labels unchanged) and training models solely on this synthetic data.

The potential privacy risks in this approach include: (1) the final model could still memorize mislabeled data, as the labels are retained, and (2) the fine-tuned diffusion model, having been trained on private data, may leak sensitive information.

The first privacy risk could be mitigated by using text-to-image generation, which does not require labels and is thus less sensitive to mislabeled canaries. In Figure 10, while image-to-image generation still leaks privacy, text-to-image generation results in predictions close to random guessing.

Regarding the second privacy risk, since stable diffusion is pretrained on large public datasets and has strong generative capabilities, it is largely insensitive to mislabeled data and does not exhibit strong memorization. As seen in Figure 10, when using text-to-image generation, the only privacy leakage stems from the fine-tuning process. However, based on membership inference success, this fine-tuning does not result in significant privacy leakage.

A more interesting direction for future research would be to investigate the privacy-utility tradeoff in diffusion models that are trained from scratch. This approach would enable a fairer comparison with other methods that do not rely on large-scale pretraining datasets.

Table 4: Fair comparison with DPSGD. We compare the privacy-utility-efficiency tradeoff of the heuristic defenses we study to two DPSGD baselines tuned for different test accuracy.

| Method | Test Accuracy | TPR@0.1%FPR | Efficiency |
|---|---|---|---|
| Undefended | 94.78% | 100.00% | 1.0× |
| Coreset Selection | 92.57% | 72.94% | 2.9× |
| MTT (OOD) | 80.42% | 0.18% | 39.7× |
| DFKD | 92.09% | 43.38% | 2.9× |
| Diffusion | 82.64% | 12.65% | 385.4× |
| DPSGD (medium) | 83.26% | 0.40% | 0.4× |
| DPSGD (high) | 90.36% | 9.31% | 1.9× |

## 4.6 FAIR COMPARISON WITH DPSGD

Since differential privacy is the most standard defense, we provide a fair comparison of all four training paradigms against DPSGD (Abadi et al., 2016). We tune each method to achieve comparable test accuracy and then evaluate both privacy leakage and training efficiency. Specifically, we consider two DPSGD baselines: one with a test accuracy of 83.26%, and a high-utility baseline achieving 90.36% in accuracy.

We present all results in Table 4, where it is clear that DPSGD remains the best defense, achieving a superior trade-off between privacy, utility, and efficiency. It is worth noting that methods like DD and fine-tuning diffusion models are significantly more time-consuming, requiring substantially more training hours compared to DPSGD.

## 5 CONCLUSION

Empirical methods that claim to preserve data privacy, whether implicitly or explicitly, but lack theoretical guarantees (e.g., differential privacy), require careful scrutiny. In this work, we systematically and rigorously evaluate privacy leakage across all four training paradigms that rely on synthetic data. To avoid providing a false sense of security, we consistently report privacy leakage in the worst-case scenario and conduct fair comparisons with a differential privacy baseline. Our results show that none of these empirical methods achieve a better trade-off than DPSGD.

We hope this work can help researchers gain a deeper understanding of privacy, and that any new empirical defenses should also undergo rigorous evaluation.

## REPRODUCIBILITY STATEMENT

To ensure the reproducibility of our results, we provide a detailed description of our attack protocols, as well as the hyperparameters and training details for all defenses in Section 4.1 and Appendix A. The source code is available at `https://github.com/yunpeng-zhao/syndata-privacy`.

## ACKNOWLEDGMENTS

J.Z. is funded by the Swiss National Science Foundation (SNSF) project grant 214838. We sincerely thank Michael Aerni and Bo Zhao for their insightful discussions during the early stages of our experiments on dataset distillation, as well as the anonymous reviewers for their valuable feedback and constructive suggestions.

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

# A TRAINING DETAILS

## A.1 UNDEFENDED

For the undefended baseline, we employ the same training procedure as described in (Aerni et al., 2024). Concretely, ResNet-18 models are trained using the SGD optimizer with a momentum of 0.9 and a weight decay of 0.0005. We use a batch size of 256 and typical data augmentation techniques, including random horizontal flips and random shifts of up to 4 pixels. The models are optimized over 200 epochs with a base learning rate of 0.1. We employ a linear warm-up of the learning rate during the first epoch, followed by a decay of the learning rate by a factor of 0.2 at epochs 60, 120, and 160.

## A.2 CORESET SELECTION

We adjust the coreset size to balance the privacy-utility-efficiency tradeoff, which is set to 20,000 to maintain high utility while using as few samples as possible. Reducing the coreset size further results in a significant drop in model test accuracy. The training process and other hyper-parameters for coreset selection remains consistent with the original paper (Toneva et al., 2019). The resulting coresets are used to train the corresponding shadow models in the same way as the undefended baseline in Appendix A.1.

## A.3 DATASET DISTILLATION

Given that we consistently use ResNet-18 as the network architecture for our shadow models across various defenses, we initially considered employing it for DD as well. However, in the high image-per-class (ipc) setting, which is essential for improved utility, we faced challenges with the DD process. The optimization had difficulties converging, ultimately failing to deliver satisfactory results. Consequently, we opted for the ConvNet architecture specified in (Zhao & Bilen, 2023), which includes three repeated convolutional blocks. Each block comprises a 128-kernel convolution layer, an instance normalization layer, a ReLU activation function, and an average pooling layer.

Studies in (Guo et al., 2024) have shown that the synthetic dataset generated through DD exhibits satisfactory cross-architecture performance, especially with a large ipc. This implies that using ConvNet for DD does not result in significant performance discrepancies when training shadow models compared to using ResNet-18. To maximize utility, we set the ipc for all DD methods to 1,000. Upon generating the synthetic data, we follow the method described in Appendix A.1 to train our shadow models with the ResNet-18 architecture.

For each DD defense, we explore three different methods for initializing the synthetic dataset: Private, Noise, and OOD initialization. Private initialization involves randomly sampling images class-by-class from the original private training set, and Noise initialization employs Gaussian noise directly. Private initialization is commonly used in practice, while Noise initialization is less discussed due to its challenging optimization process and suboptimal performance. In this work, we attempt to use OOD data for initialization. For example, we use CINIC-10 (Darlow et al., 2018), an extension of CIFAR-10 incorporating downsampled ImageNet images, for initialization. For each class, we select 1,000 images from CINIC-10 that corresponded to the classes in CIFAR-10 but are not included in it. Next we will introduce the specific implementation details for each DD defense.

**DM & DSA** For DM and DSA defenses, it is crucial to adjust the learning rate and training iterations based on different synthetic dataset initialization methods. DSA, involving a dual-layer optimization process, additionally requires specification of the number of both the outer and inner loops. Table 5 provides the hyperparameter settings for DM and DSA. All other training settings align with those outlined in their original publications (Zhao & Bilen, 2023; 2021).

Table 5: Hyperparameters used for DM and DSA. Private, Noise, and OOD are initialization methods. LR indicates the learning rate. Outer and Inner refer to the number of outer and inner loops.

| Methods | LR | Iteration | Outer | Inner |
|---------|-----|-----------|-------|-------|
| DM (Private) | 10 | 30,000 | - | - |
| DM (Noise) | 50 | 100,000 | - | - |
| DM (OOD) | 30 | 100,000 | - | - |
| DSA (Private) | 0.1 | 300 | 100 | 1 |

Table 6: Hyperparameters used for the synthetic dataset optimization process in MTT and DATM.

| Methods | Iteration | Synthetic Steps | Expert Epochs | Min Start Epoch | Max Start Epoch | Synthetic Batch Size | LR (Pixels) | LR (Labels) | LR (Step Size) | Starting Step Size |
|---|---|---|---|---|---|---|---|---|---|---|
| MTT (Private) | 1,500 | 100 | 2 | - | 40 | 200 | 100 | - | 1e-6 | 0.01 |
| MTT (OOD) | 20,000 | 100 | 2 | - | 40 | 200 | 100 | - | 1e-6 | 0.01 |
| DATM (Private) | 5,000 | 100 | 2 | 40 | 60 | 1,000 | 50 | 10 | 1e-6 | 0.01 |

**MTT & DATM** For MTT and DATM, training of the teacher trajectory precedes the optimization of synthetic data. We extend the trajectory length for MTT to 100 epochs while keeping other hyperparameters and settings consistent with those described in the original papers (Cazenavette et al., 2022; Guo et al., 2024). Table 6 shows the hyperparameters during the synthetic dataset optimization process. All other training parameters remain in alignment with those specified in the respective original methods. ZCA whitening is used in all experiments for MTT and DATM by default. Considering the heavy reliance of MTT and DATM on GPU memory, we adopted the TESLA (Cui et al., 2023) implementation of DATM and also re-implemented MTT as a TESLA version, following recommendations in (Guo et al., 2024).

## A.4 DATA-FREE KNOWLEDGE DISTILLATION

In line with the protocol described in (Fang et al., 2022), using only the "BN" loss is able to achieve high accuracy. This loss function matches the batch-normalization statistics from the teacher model, which is widely used in most DFKD works. Given its effectiveness, we use this method exclusively to generate synthetic data. This simplifies our approach and allows us to generalize our evaluation to other DFKD methods.

The training protocol in Appendix A.1 is adopted to train the teacher model. After that, we perform the distillation process involving 240 iterations. In each iteration, we generate 256 new images, collect predictions from the teacher model, and store these in a memory bank. We then train the student model for 5 epochs using the data in the memory bank.

## A.5 DIFFUSION MODEL

Due to the substantial computational costs associated with Real-Fake (Yuan et al., 2024), we have made slight adjustments to our experimental setup. In all experiments involving diffusion, we reduced the number of sampling steps during the inference stage to 10. Furthermore, we decreased the number of shadow models from 32 to 16. These modifications allow us to reduce computational expenses while still obtaining meaningful conclusions. We only generate synthetic dataset equal in size to the original private dataset. The training process of shadow models is the same as in Appendix A.1.

## A.6 DP-SGD

The two heuristic DPSGD baselines without provable privacy guarantees in Table 4 are developed following (Aerni et al., 2024), building upon the state-of-the-art DPSGD training techniques (De et al., 2022; Sander et al., 2023). Here, we replace the batch normalization in ResNet-18 with group normalization (Wu & He, 2018), swap the order of normalization and ReLU. We carefully tune the hyperparameters according to the scaling law in (Sander et al., 2023) to achieve medium utility and high utility (see Table 7). Finally, the privacy budgets are $\epsilon \approx 1.8 \times 10^8$ for high-utility DPSGD and $\epsilon \approx 4.4 \times 10^7$ for medium-utility DPSGD.

Table 7: Hyperparameters of heuristic DPSGD baselines with medium utility and high utility.

| Method | Noise multiplier | Clipping norm | Batch size | Epochs | LR | Augment |
|---|---|---|---|---|---|---|
| DPSGD (medium) | 0.00625 | 1 | 64 | 4 | 4 | 8 |
| DPSGD (high) | 0.00625 | 1 | 64 | 16 | 4 | 8 |

# B  Additional Experiments and Analysis

## B.1  Privacy Leakage in DD with Private Initialization

When using Private initialization, a portion of mislabeled canaries is inadvertently selected as the starting images for synthesizing categories, e.g., a cat mistakenly labeled as a dog could be used for initializing dog category images. As discussed in (Guo et al., 2024), under high ipc settings, the images optimized through DD remain visually similar to the original images. It is important to emphasize that visual similarity should not be deemed the determinant of privacy leakage; however, such similarity does indicate potential privacy risks.

We define True Positive samples at an FPR of 0.1% as those successfully attacked, and divide the training set's canaries into init canaries and non-init canaries based on their selection for initialization. Table 3 shows the attack success rates for these two categories under different DD defenses. It is evident that the success rate on init canaries is significantly higher than on non-init canaries. For canaries in the training set but not used for initialization, the success rate is nearly equivalent to random guessing. This empirically supports our assertion that privacy risks are predominantly introduced during the initialization phase rather than the distillation process.

## B.2  Privacy Leakage During the Distillation Process in DD

We then perform a case-by-case analysis to explain why the distillation processes of these DD defenses do not leak the membership privacy of mislabeled canaries.

DM matches the means of neural network-encoded representations of real and synthetic data class by class. During the optimization of synthetic data, the neural network is randomly sampled from the parameter space and undergoes only forward propagation without updating its parameters, thus avoiding any memorization effects related to mislabeled canaries. Given that mislabeled samples constitute a minor fraction of the training data (about 1%), their influence on the average representation of the original real data is negligible. Therefore, the update process of synthetic data is virtually unaffected by the canaries.

However, defenses such as DSA, MTT, and DATM initially require training models on original private data to obtain teacher gradients or parameters. Subsequently, models are trained from the same starting point using synthetic data, aiming to elicit similar parameter updates from both real and synthetic data. The privacy risk here stems from the potential retention of canary-related memories in the gradient or parameter trajectories during the teacher model's training process. The optimization goal previously mentioned could lead to models trained on the final synthetic dataset inheriting strong memories of the canaries.

Fortunately, we find that model memories of mislabeled canaries do not form rapidly (see Figure 11). In early training epochs, the model have not formed strong memorization to those mislabeled canaries. This finding may provide a reasonable explanation for why there is no privacy leakage during the synthetic data optimization processes in DSA, MTT, and DATM.

DSA randomly samples a neural network at each optimization epoch and trains it incrementally with both synthetic and real data, minimizing the difference between their gradients to update the synthetic images. Since the number of training steps is generally small, the teacher gradients do not yet contain effective memories of the canaries.

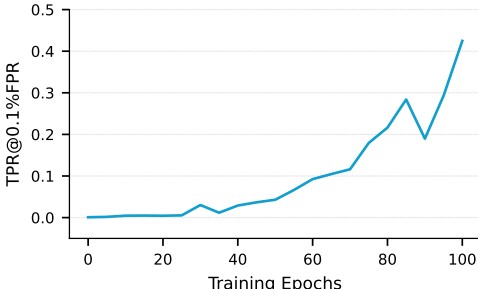

Figure 11: The TPR@0.1%FPR of 500 mislabeled canaries during the training process of teacher trajectories.

Trajectory-based methods like MTT and DATM first train a teacher model on original private data and save the training trajectory. In each optimization epoch, a network is selected from the teacher trajectory and trained for several steps with synthetic data to match the trained model parameters to the teacher trajectory. However, both MTT and DATM match only the early to mid-training tra-

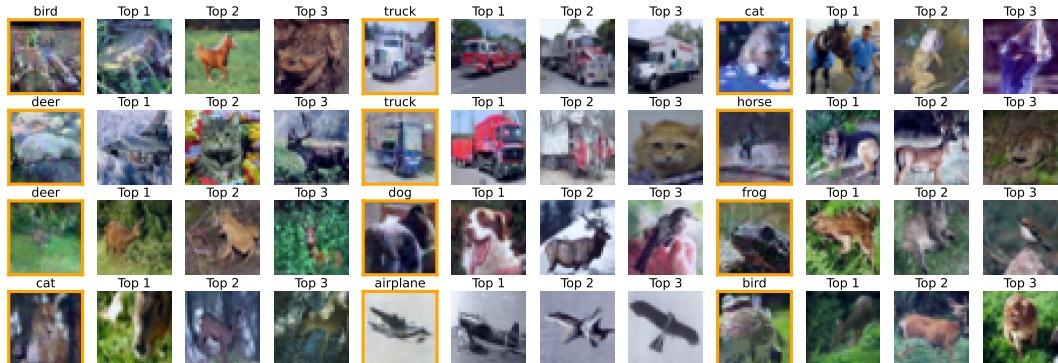

Figure 12: Visualization on synthetic data generated by DM (OOD) and retrieved private data with SSCD. The synthetic data are with orange border. The synthesized data does not exhibit evident visual similarity to any of the private data.

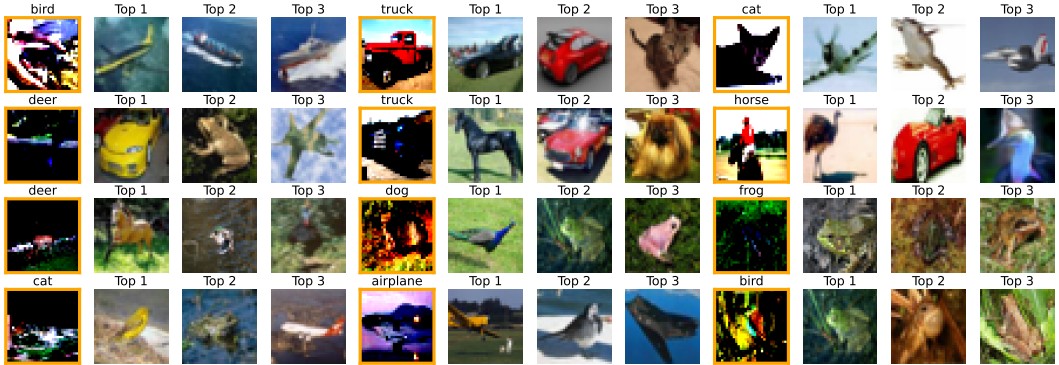

Figure 13: Visualization on synthetic data generated by MTT (OOD) and retrieved private data with SSCD. The synthetic data are with orange border. The synthesized data does not exhibit evident visual similarity to any of the private data.

jectories, during which neither the model parameters in the teacher trajectory have formed strong memories of the canaries. Hence, these gradient and trajectory-matching optimization processes prevent models trained on synthetic data from retaining memories of the canaries, thereby safeguarding member privacy. This also explains why using noise and OOD images for initialization can achieve decent privacy protection.

### B.3 Visual Dissimilarity Between Private Data and Data Synthesized by DD with OOD Initialization

Following the previous works (Somepalli et al., 2023; Guo et al., 2024), we employ the Self-Supervised Content Duplication (SSCD) (Pizzi et al., 2022) method for content plagiarism detection. We use the ResNet50 model trained on the DISC dataset (Douze et al., 2021). For a given query synthetic image and all reference images from the original private dataset, we infer through the detection model to obtain a 512-dimensional feature vector for each image. The similarity between them is calculated by computing the inner product between the query feature and each reference feature. For each synthetic image, we present the corresponding private images with top 3 similarities.

We select some synthetic images from DM and MTT, both with OOD initialization, and attempt to retrieve similar images from the original private dataset. The results are shown in Figure 12 and Figure 13, respectively. It is obvious that for DD methods with OOD initialization, the synthetic data is visually dissimilar to those private data.

### B.4 DETAILED ANALYSIS OF THE PRIVACY LEAKAGE IN DFKD

In this section, we investigate how membership information of mislabeled canaries is transferred from the teacher to the student model through the use of dissimilar synthetic data. As demonstrated in Figure 9, the teacher model can strongly memorize the incorrect labels associated with mislabeled samples, resulting in a logits distribution where the value of wrongly-labeled class significantly exceed that of the correct labels. The generator in DFKD is capable of producing synthetic images that, while visually distinct, bear a strong resemblance in logits to the mislabeled data. These synthetic images activate the memories of corresponding canaries encoded in the teacher model's parameters. The distillation process attempts to align the student model's logits distribution with that of the teacher for the same images. Consequently, the student model may inherit similar parameters to the teacher, including some aspects of the teacher's memorization of mislabeled data.

### B.5 DETAILED ANALYSIS OF FIGURE 9

Figure 9 reveals that, despite having no direct exposure to the original mislabeled data, the student model still inherits a portion of the teacher model's memory, exhibiting high logit value for the mislabel. Notably, the student model's memorization of mislabels is comparatively weaker than teacher. Therefore, influenced by its generalization capabilities, the student model tends to classify canaries under their correct labels rather than the mislabels.

We assessed the accuracy with which both teacher and student models classify the training set's canaries into their respective incorrect labels. The results were 100% for the teacher and 5.28% for the student, indicating that using accuracy as the metric to evaluate whether a model remembers mislabels is misleading. Even if the logits for a mislabeled class are not the highest, their magnitude may still suffice for effective LiRA.

## C EVALUATION SETUP OF FIGURE 3 AND FIGURE 4

For Figure 3, the experiments were conducted on CIFAR-10, where we used 500 mislabeled samples to simulate the most vulnerable data. The attack settings follow those of LiRA. For each defense, we trained 16 models, ensuring that each sample appeared in the training set of only half of the models. We used leave-one-out cross-validation—each time using one model as the victim and the remaining 15 models as the attacker's shadow models. The shadow models were implemented using a ConvNet architecture. We evaluated privacy protection levels for both the average case and the worst case.

Figure 4 follows the same experimental setting as the undefended baseline without any defense, described in Appendix A.1, except that we set the number of shadow models to 256 instead of 32. For the "average-case" subfigure, we selected a normal sample and plotted its loss distributions when it was a member and when it was not. For the "most vulnerable" subfigure, we selected a mislabeled canary and did the same.

