# OpenReview forum: "Does Training with Synthetic Data Truly Protect Privacy?"
_ICLR.cc/2025/Conference — ICLR 2025 Poster_

### Official Review · Reviewer_YLFf · 2024-10-25

**Soundness:** 3
**Presentation:** 2
**Contribution:** 3
**Rating:** 6
**Confidence:** 4

**Summary:**

This paper provides a systematic analysis for privacy risk of the usage of synthetic data. The investigated methods include core set selection, dataset distillation, data-free distillation and diffusion models. The estimated privacy leakage is based on 500 canaries that are mislabeled and LiRA by shadow models. The results show that the privacy risk still remains and higher than DP-SGD at the same comparison of utility. This work also provides several detail analysis including the logits similarity.

**Strengths:**

- This work presents a systematic evaluation of privacy metrics TPR-FPR by LiRA for four synthetic data generation methods, and show that privacy leakage of synthetic data is much higher in than the results in previous literature.
- This work also provide some detailed analysis including considering visually similarity and logits similarity together instead of relying on the single MIA metric.

**Weaknesses:**

- The analysis could be improved (see questions below).
- The technical contribution is somewhat limited. For example, the privacy estimation metric is based on previous work LiRA.

**Questions:**

1. It is interesting to see the analysis for visual similarity and logit similarity. I did not quite follow the analysis for why the logit similarity of the synthetic data in DFKD leads to the higher MIA metric of private data. Figure 9 shows that the logit of the private data in student model prediction is much different than the remaining three, therefore I wonder if the authors could explain why this logits similarity of the synthetic data could lead to the higher MIA metric.

2. Figure 7 shows that private data that has high confidence in original label and Figure 9 shows the private data that has higher confidence in the mislabel label. Would this show the different canary property for different kind of methods.

3. The dataset distillation method (random/OOD) shows utility drop compared to DP-SGD. Would increasing the number of synthetic data help improve utility?

4. It seems to me the TPR value granularity for 500 canaries is 0.2% (1/500) and I wonder how is the TPR reported in Tables are calculated. Or in other words, the authors may provide a detailed description about how MIA metric is calculated.

5. I wonder if the author could provide an adaptive attack that achieves higher attacker performance based on Figure 7 findings where the LiRA achieves a lower MIA metric.

I am happy to increase my score if I find my concerns addressed.

---

> ### Author Response · Authors · 2024-11-18
> **Reply to Reviewer YLFf [1 / 2]**
>
> Dear Reviewer YLFf:
>
> We sincerely appreciate your recognition of our systematic evaluation and find our analysis of visual similarity and logits similarity interesting. We now address your concerns below.
>
> ---
>
> > Q1: The technical contribution can be somewhat limited.
>
> We would like to kindly clarify that we are not claiming our contribution as proposing a new or stronger MIA method. Our aim is to show that even with existing MIA methods, it becomes evident that there are numerous misleading claims regarding the use of synthetic data in training models.
>
> Breaking the four defenses we analyzed was not particularly challenging. This highlights that before researchers claim their empirical defenses to be privacy-preserving, they should take one additional step - conducting rigorous evaluations, as demonstrated in our work. Such evaluations are not difficult to perform. We hope this inspires future researchers to approach this issue more carefully, which is our primary goal.
>
> > Q2: Why synthetic data with similar teacher logits as the original private data harms the privacy of private data and leads to higher MIA metrics.
>
> That’s a great question! We’ll answer it from the following two perspectives:
>
> - Why previous studies suggested that DFKD can protect privacy: Intuitively, since the student model is never trained on private data, it is assumed that releasing the student model should safeguard the privacy of private data.
>
> - Why DFKD actually **fails** to protect privacy: For the most vulnerable data point in the dataset (denoted as X), the teacher model memorizes this sample strongly. During the distillation process, a synthetic data point X’ (which is **visually completely dissimilar** to X) may inadvertently trigger the teacher model's memory of X. In the MIA process, if the logits similarity between X and X’ is very high, it implies that during the MI attack, even though the student model never directly saw private data X, it still indirectly learned about X by seeing synthetic data X’.  As a result, the membership privacy of X is completely leaked.
>
> Therefore, even if a model has never directly seen private data, it can still potentially leak privacy. We should not rely on data that is visually completely dissimilar to private data as a guarantee of privacy protection. More rigorous evaluations are essential.
>
> > Q3: Figure 7 shows that private data that has high confidence in original label and Figure 9 shows the private data that has higher confidence in the mislabel label. Would this show the different canary property for different kind of methods.
>
> That's a great observation! Indeed, canaries should be carefully tailored to different defenses and datasets. For instance, mislabeled data would not be effective canaries for defenses based on self-supervised learning, which does not rely on labels.
>
> In our work, we did not use the most optimized canaries for each defense we evaluated. This means that with better-designed canaries, it's likely that even higher MIA success rates could be achieved. Despite this, our approach of using mislabeled data as canaries already yielded strong results, clearly demonstrating that these defenses provide a false sense of privacy protection.

---

> > ### Author Response · Authors · 2024-11-18
> > **Reply to Reviewer YLFf [2 / 2]**
> >
> > > Q4: The dataset distillation method (random/OOD) shows utility drop compared to DP-SGD. Would increasing the number of synthetic data help improve utility?
> >
> > Yes, increasing the number of synthetic data points could potentially improve utility, but this goes against the original motivation of DD. The goal of DD is to achieve similar performance to the full dataset while using significantly fewer data points (e.g., ipc=10 or 100).
> >
> > Moreover, when ipc = 1000, each experiment consumes approximately 563 GPU hours (nearly 100 times the runtime of DPSGD). Since increasing IPC further does not enhance privacy, it may not be worthwhile.
> >
> > Additionally, related works rarely use OOD data for dataset distillation. We encourage researchers to explore this direction!
> >
> >
> > > Q5: More details about how the MIA metric is calculated.
> >
> > Thank you for pointing this out.
> >
> > As we have mentioned in the experimental setup, we randomly designate 500 data points as “audit samples” on which we evaluate membership inference, and we use mislabeled data as strong canaries to simulate worst-case data; the remaining 49,500 samples are always included in every model's training data.
> >
> > For each method, we train 32 shadow models, ensuring that each audit sample is included in the training data of 16 models. Here, we used leave-one-out cross-validation for all 32 models—each time using one model as the victim and the remaining 31 models as the attacker's shadow models. Therefore, we can calculate the attack’s TPR and FPR over the 32×500 guesses of the attacker on all canaries and victim models. We will update this to the experimental setup part in the revised manuscript.
> >
> > > Q6: Could provide an adaptive attack that achieves higher attacher performance based on Figure 7 findings where the LiRA achieves a lower MIA metric.
> >
> > We really appreciate your suggestion on designing an adaptive attack. Actually, we put a lot of effort into this, but we find that it is really challenging to represent the “weird” DD transformation with a simple function. Unlike other common augmentations—such as cropping, rotation, or Gaussian blur, which are relatively straightforward to model—DD transformation is much harder to represent, making adaptive attacks more complex to implement.

---

> ### Comment · Reviewer_YLFf · 2024-11-25
>
> Thank you for your detailed response.
>
> Q2. I am still somewhat confused about privacy leakage about the logits similarity between teacher's prediction on private image, teacher's prediction on synthetic image, and student's prediction on synthetic image.
>
> Figure 9 shows that the student's prediction on the private image is different than the other three and therefore it is not clear to me what there is a privacy concern for the private image regards to the student model.
>
> Q4 &Q5. thanks for the clarification.
>
> Q6. thanks for your efforts.
>
> For Q1, along with Q2, Q3, Q6, given that there are already prior works showing that evaluations of privacy preserving machine learning methods should consider rigorous evaluation, and the dataset distillation method studied in this work, its privacy leakage is also studied in [2], I wonder if this work could provide some outlines and suggestions how to to design the rigorous evaluations for future research.
>
> [1] Aerni et al. Evaluations of Machine Learning Privacy Defenses are Misleading. CCS 2024.
>
> [2] Carlini et al. No free lunch in” privacy for free: How does dataset condensation help privacy. arxiv 2022

---

> > ### Author Response · Authors · 2024-11-25
> > **reply**
> >
> > Thanks for your reply!
> >
> > > why there is a privacy concern for the private image regards to the student model.
> >
> > Simply put, when trained on data $x$, as long as the student model also has a relatively high probability on the canary label $y'$ (and the teacher model has a very high probability on this label, as shown in Fig 9.), this means the student model will have a very low loss, e.g., cross_entropy(f(x), y') is very low; when the teacher model is not trained on x, cross_entropy(f(x), y') is very high (due to model generalization, the model should predict it as true label $y$). This huge gap in losses provides a strong membership signal.
> >
> > > provide some outlines and suggestions how to to design the rigorous evaluations for future research.
> >
> > Before claiming that an empirical defense really protects privacy, make sure to try an adaptive attack first. Look into the privacy leakage in the worst-case scenario and, if necessary, the average case as well. Then do a fair comparison with DPSGD.  Usually, this gives a solid evaluation overall.
> >
> > Another tip is to visualize the synthetic data, like we've shown in some dataset distillation methods. If the synthetic data looks just like the private data, then the defense is certainly not a good defense. However, visual dissimilarity does not ensure that privacy is preserved.

---

> > > ### Comment · Reviewer_YLFf · 2024-11-26
> > >
> > > Thank you for your reply, that has addressed most of my concerns. I have raised my score to 6.

---

### Official Review · Reviewer_ybiF · 2024-10-27

**Soundness:** 2
**Presentation:** 3
**Contribution:** 3
**Rating:** 6
**Confidence:** 2

**Summary:**

This paper investigates whether using synthetic data in machine learning genuinely safeguards privacy, as often claimed. The evaluations is done on 4 different training paradigms-coreset selection, data distillation, data-free knowledge distillation and synthetic data generated from diffusion models. To test the privacy claims of these methods, the study uses membership inference attacks (MIAs), focusing on worst-case scenarios to rigorously assess privacy leakage. The paper also compares these methods to Differential Privacy Stochastic Gradient Descent (DPSGD), a technique known for providing formal privacy guarantees, and finds that DPSGD consistently outperforms synthetic data-based approaches in terms of the privacy-utility-efficiency balance. The findings reveal that none of the synthetic data techniques match DPSGD in safeguarding privacy effectively. Notably, the study also discovers that visual dissimilarity between synthetic and private data does not necessarily imply privacy protection, as even visually distinct synthetic data can leak information when model logits are similar. This highlights a risk that methods relying solely on visual or distributional differences may offer a false sense of privacy.

**Strengths:**

This broad approach offers a thorough understanding of various methodologies in synthetic data utilization and their impact on privacy.
The study juxtaposes synthetic data-based techniques with Differential Privacy-SGD (DPSGD) as a baseline, which helps readers contextualize the efficacy of synthetic data methods in privacy preservation compared to a gold-standard approach like DPSGD.
The study identifies instances where synthetic data, despite visual dissimilarity from private data, can still leak privacy information through logit similarities. This nuanced finding enhances the paper's depth by showing that visual similarity alone isn’t sufficient to evaluate privacy.

**Weaknesses:**

The experiments focus on CIFAR-10 and specific models, such as ResNet-18, which may limit the generalizability of findings. The paper’s findings could vary across more complex datasets or architectures, and broader experiments could better represent the implications for privacy in diverse real-world scenarios​.

Techniques like DPSGD are noted for efficiency, yet they are resource-intensive. The paper briefly mentions but does not deeply engage with the practical constraints of computational cost and scalability, which are critical factors for real-world implementation of privacy-preserving methods​.

While the empirical evaluation is thorough, the paper lacks an in-depth theoretical framework to explain why certain synthetic data techniques lead to privacy leakage. A theoretical grounding could bolster the empirical findings and offer predictive insights for synthetic data privacy.

**Questions:**

Why do you consider coreset selection as synthetic data?

---

> ### Author Response · Authors · 2024-11-18
> **Reply to Reviewer ybiF**
>
> Dear Reviewer ybiF,
>
> Thank you so much for your thoughtful feedback and for taking the time to review our work—we truly appreciate it!
>
> We understand your concern about our experiment being conducted on a single dataset and the lack of theoretical support (please refer to our detailed response below). That said, we’d like to emphasize that many studies in privacy evaluation often present misleading conclusions, and our experiment serves to highlight this critical issue. Through our work, we aim to encourage researchers to adopt more rigorous evaluation practices before claiming their methods are privacy-preserving—a step we believe is incredibly important!
>
> Tackling systemic issues in research methodologies, especially in evaluations, is central to advancing the field. We respectfully ask you to reconsider the potential impact of our contribution, as it has the ability to positively shape future research directions.
>
> Once again, thank you for your valuable feedback. We address your concerns in detail below.
>
> ---
>
> > Q1: Experiments on more datasets and more network architectures.
>
> Thank you for suggesting reporting results on more datasets and network architectures. While we agree that using more datasets would nevertheless have been interesting, we decided against this:
>
> (1) To do fair comparisons, we needed all studied methods to achieve reasonable test performance. Some methods, such as Dataset Distillation, perform poorly on more complex datasets like CIFAR-100 or ImageNet, which would make comparisons less meaningful.
>
> (2) Similar datasets (e.g., CIFAR-100) Figure 1 alone requires more than 5800 GPU hours (without hyperparameters tuning and additional experiments). We decided that the auxiliary insights were not worth the cost.
>
> (3) Our approach aligns with previous studies like [1], which also conducted comprehensive evaluations of defense methods primarily on CIFAR-10. This consistency facilitates a direct comparison between our results and existing literature.
>
> As our goal is to reveal the false sense of privacy protection of existing methods that train ML models with synthetic data, we believe that it is sufficient to focus on the most standard dataset used in empirical evaluations of privacy defense.
>
> > Q2: DP-SGD is resource-intensive. Discussion about computational cost and scalability in real-world scenarios is required.
>
> Our main argument is that DP-SGD can serve as a strong heuristic defense, providing both good utility and empirical privacy protection. In fact, when using a small noise level and setting the privacy budget $\epsilon \gg 1000$,   DP-SGD can be more computationally efficient than many other defenses.
>
> The key insight is that, by using small noise, we forgo DP-SGD’s provable privacy guarantee in exchange for significantly higher utility, while also improving efficiency.
> For training a single model with high-utility DP-SGD in our experiments, it only took **78** minutes. In this way, it is not particularly resource-intensive.
>
>
>
>
> > Q3: An in-depth theoretical framework to explain why certain synthetic data techniques lead to privacy leakage.
>
> Thank you for the valuable suggestion. We agree that a theoretical framework is indeed helpful and can offer predictive insights. However, since our focus is primarily on different types of **heuristic** defenses, which are highly heterogeneous, we believe it is both extremely difficult and not very meaningful to provide a generic theoretical framework for these defenses.
>
> Additionally, none of these methods come with a theoretical foundation themselves, as they are heuristic defenses rather than theoretical defenses.
>
>
>
> > Q4: Why is the data from coreset selection considered as synthetic data?
>
> Thank you for this question! Our initial idea is that, although coreset selection does not involve a data synthesis process, it shares a common goal with the other synthetic data methods: obtaining an informative proxy training set. The difference lies in the approach—data synthesis methods generate samples directly, while coreset selection automatically selects an informative subset from the original dataset. Therefore, we gave a footnote on page 1 to explain that, for simplicity, we use the term “synthetic data” (also for coreset) in the rest of the paper.
>
> Moreover, we would like to note that coreset selection just serves as a starting point—a very simple method—to demonstrate that average-case evaluations can be misleading. Additionally, through studying coreset selection, there is an interesting finding that the selection or unlearning of specific samples could introduce further privacy leakage—some selected samples exhibit a greater degree of privacy leakage compared to when they are part of the entire training set.
>
> We will also consider moving this discussion to the appendix in the revised manuscript.
>
>
> [1] "Evaluations of Machine Learning Privacy Defenses are Misleading." CCS 2024.

---

> > ### Comment · Reviewer_ybiF · 2024-11-25
> > **Thank you for your responses**
> >
> > I have read your responses and have decided to increase my score.

---

### Official Review · Reviewer_cuks · 2024-11-03

**Soundness:** 2
**Presentation:** 4
**Contribution:** 3
**Rating:** 6
**Confidence:** 3

**Summary:**

This paper measures empirical privacy for several methods of training vision models from synthetic data that claim to preserve privacy to some degree with respect to the training data: CoreSet selection, Dataset Distillation, Date-Free Knowledge Distillation, and synthetic data from Fine-Tuned Diffusion models. Empirical privacy is evaluated using the Likelihood Ratio Attack (LiRA) and the general setup follows [0]. As a private baseline, DP-SGD is used.

For the evaluation, a ResNet architecture is used with the CIFAR-10 dataset. For each of these methods, the authors measure privacy leakage (true positive rate given a fixed false positive rate), utility (test set accuracy), and efficiency (# of training hours). The main findings are that there's a clear privacy-utility tradeoff among methods once privacy is considered as a worst-case rather than an average case notion and that none of the synthetic data methods outperform DP-SGD with the three metrics jointly considered.

**Strengths:**

- This paper provides an apples-to-apples empirical privacy comparison for several methods of training vision models that claim to preserve privacy to some degree. In doing so, it improves upon prior empirical privacy analyses, some that were rather flawed.
- The presentation is clear and the text is well-written. The figures throughout are particularly helpful. See question 1 on this point.

**Weaknesses:**

- The DP-SGD comparison is potentially misleading. The "baseline" method does not satisfy differential privacy. See question 2. The paper would benefit from including an additional baseline that satisfies DP e.g. [4] discusses how to incorporate hyperparameter tuning into the privacy analysis. It may also be interesting to include a fully non-private baseline using a standard training routine i.e. just train ResNet with SGD.
- The paper would benefit from a discussion of how the formal guarantee of differential privacy differs from heuristic privacy - particularly, with respect to DP applying to adding or removing any possible possible record within the data domain; whereas, the MI attacks in this paper are only considered with respect to the training data.

**Questions:**

1. Why are the gridlines in Figure 1 not uniform? Consider changing this or explaining in figure caption.
2. As with [0], are the DP-SGD models reported in Table 4 only non-private due to hyperparameter tuning? If so, what privacy parameters were these models trained with?

Additional stylistic notes:
- Odd phrasing on Line 83: "none of these fancy methods with synthetic data"

- Typo on Line 95: "it can provide a decent privacy protection"

- Odd phrasing Line 244: "but not evaluated in the right way"

Additional suggestions:
- When discussing the intuition for why initialization on public data provides privacy guarantees, consider referencing [1, 2, 3], which illustrate how public data can be used to improve differentially private synthetic data.

[0] Aerni, Michael et al. "Evaluations of Machine Learning Privacy Defenses are Misleading." 2024.

[1] Liu, Terrance, et al. "Leveraging public data for practical private query release." International Conference on Machine Learning. PMLR, 2021.

[2] Liu, Terrance, et al. "Iterative methods for private synthetic data: Unifying framework and new methods." Advances in Neural Information Processing Systems 34 (2021): 690-702.

[3] Fuentes, Miguel, et al. "Joint Selection: Adaptively Incorporating Public Information for Private Synthetic Data." International Conference on Artificial Intelligence and Statistics. PMLR, 2024.

[4] Papernot, Nicolas, and Thomas Steinke. "Hyperparameter tuning with renyi differential privacy." ICLR, 2022.

---

> ### Author Response · Authors · 2024-11-18
> **Reply to Reviewer cuks**
>
> Dear Reviewer cuks:
>
> Thank you for your insightful comments on our paper. We sincerely appreciate your recognition of how our work advances prior analyses of empirical defenses using synthetic training data. We now address your concerns below:
>
> ---
>
>
> > Q1: Comparison with DP-SGD baselines could be misleading as they do not satisfy differential privacy. Could add a baseline that satisfies DP and a non-private baseline.
>
> Thanks for the valuable suggestions and we are sorry for anywhere that is unclear.
>
> It is true that we give up meaningful provable privacy guarantees and view DP-SGD as a purely heuristic defense. This is exactly our point because we need to tune the hyper-parameters of these DP-SGD baselines for higher utilities to ensure fair comparisons with other empirical defenses. Results prove that none of the empirical defenses based on synthetic training data outperforms heuristic DP-SGD baselines in the context of privacy-utility-efficiency tradeoff under similar utility levels. We will improve the writing to clarify and avoid being misleading.
>
> For the non-private baseline, it is reported in Figure 1 and Table 4 named “Undefended”. It represents a ResNet trained with a standard training routine using SGD. TPR@0.1%FPR of the non-private baseline is 100% and the test accuracy is 94.78%. More details can be found in Appendix A.1. We will also improve the writing to introduce it in the main text.
>
> > Q2: The paper would benefit from a discussion of how the formal guarantee of differential privacy differs from heuristic privacy - particularly, with respect to DP applying to adding or removing any possible record within the data domain; whereas, the MI attacks in this paper are only considered with respect to the training data.
>
> Thank you for your valuable suggestions on this! We will incorporate more detailed discussions based on your input.
>
> One important point to clarify: we aim to emphasize that DPSGD can serve as a very strong heuristic defense. Unlike canonical DPSGD, where random noise is added to make the privacy budget meaningful (e.g., setting $\epsilon = 1$), our focus is on the fact that even with a privacy budget much larger than 1000, DPSGD remains one of the most effective defenses among the empirical methods we've examined.
>
>
> > Q3: Why are the gridlines in Figure 1 not uniform? Consider changing this or explaining in figure caption.
>
> Thank you for pointing out this! We will address this by updating Figure 1 in the revised manuscript to make gridlines uniform.
>
> > Q4: As with [0], are the DP-SGD models reported in Table 4 only non-private due to hyperparameter tuning? If so, what privacy parameters were these models trained with?
>
> Thanks for the question. We indeed tune the hyper-parameters to achieve higher utility (for a fair comparison with other defenses) and forgo provable privacy to construct heuristic DP-SGD defenses. The privacy budgets are $\epsilon \approx 1.8\times 10^8$ for high-utility DP-SGD and $\epsilon \approx 4.4\times 10^7$ for medium-utility DP-SGD. Their hyperparameters are provided in Appendix A.6. We apologize for not mentioning the table of privacy parameters in the appendix within the main text. We will include this in the revised manuscript to be clearer.
>
> > Q5: Additional stylistic notes and suggestions.
>
> We appreciate your careful attention to stylistic details and suggestions for improvement. For the typos and odd phrasings, we will correct them in the revised version. Regarding the intuition for why initialization on public data provides privacy guarantees, we will incorporate the suggested citations into our analysis and discussion to strengthen the argument.

---

> > ### Comment · Reviewer_cuks · 2024-11-25
> >
> > Thanks for addressing my concerns. Good paper. I'm keeping my score at 6.
> >
> > Q1: Regarding the non-private baseline, I missed that in Figure 1.
> >
> > Q4: Adding some references to the appendix will be helpful to the reader.

---

### Official Review · Reviewer_qB2Z · 2024-11-04

**Soundness:** 3
**Presentation:** 3
**Contribution:** 2
**Rating:** 6
**Confidence:** 3

**Summary:**

The paper studies the privacy-preserving properties of commonly-used synthetic image generation methods. The measuring of privacy-preservation is carried out using membership inference attacks using the attacks given by [Aerni et al. (2024)](https://arxiv.org/pdf/2404.17399). The paper compares the privacy-preservation of four commonly-used techniques: coreset selection, dataset distillation, data-free knowledge, model distillation, and synthetic data generated from diffusion models. The experiments are carried out on CIFAR-10 and the most vulnerable samples are mislabeled samples.

**Strengths:**

- The paper is very well written and the experiments are well-explained and seem solid.

- The paper can serve as a good reference for showing the strength of DP-SGD for obtaining good privacy-utility trad-eoff for synthetic image data.

**Weaknesses:**

- The paper does not truly present many novel ideas, though it is another valuable demonstration about the effectiveness of DP-SGD for obtaining good privacy-utility tradeoff for ML models when the privacy protection is measured via the most vulnerable samples.

Regarding the idea of auditing with worst-case samples, for example: it has been studied extensively and is already considered by Carlini et al, 2019, "The secret sharer: Evaluating and testing unintended memorization in neural networks" and Jagielski et al., 2020, "Auditing differentially private machine learning: How private is private sgd?" So, although auditing with worst-case samples seems to be a central theme in the paper, there is not much novelty in there (and I also think these references should be included).

Also, the vulnerability of synthetic data against membership inference attacks has been considered in the literature, and there also seem to be some central references missing: see e.g. the work by Hayes et al., 2019, "LOGAN: Membership Inference Attacks Against Generative Models."

- The experimental comparison is a bit restricted, after all, since only the CIFAR-10 dataset is considered. Perhaps one could consider datasets from other domains as well?

Despite of its strengths, I am leaning towards reject as I think the paper does not provide sufficiently novel results to reach the bar for this venue. Nevertheless, I think that with some rewriting this will be a nice paper and can serve as a reference for this topic (privacy-protection of synthetic image data).

**Questions:**

- Do you think similar comparisons could be easily carried out in different image datasets or in other domains (e.g., tabular data) ?

- What is the setting behind Figure 4? There are no details given on that experiment.

- When focussing on practical scenarios: do you think the situation would differ, if instead of using synthetic samples (e.g., mislabeled samples) for the auditing, you would try to find the most vulnerable data samples in the dataset?

- Before the experimental setup is presented on page 6, there are some experimental results presented on page 5 (Figures 3 and 4) for which no sufficient details are given. I cannot be sure for which dataset is the result of Figure 4 is, and what is exactly the setting behind that figure.

---

> ### Author Response · Authors · 2024-11-18
> **Reply to Reviewer qB2Z  [1 / 2]**
>
> Dear Reviewer qB2Z:
>
> We really appreciate your acknowledgment of the solidness of our experiments and recognition of our work as a potentially valuable reference in the topic of privacy protection for synthetic data. In the remainder of this note, we have tried to address your comments and questions in as much detail as possible. Thank you once again for your kind consideration and time. We welcome any further questions or suggestions that could improve the clarity and impact of our paper.
>
> ---
>
> > Q1: Novelty of our work. Privacy audit on worst-case samples has been extensively studied.
>
> Thanks for providing the references and we will include a discussion of them in the related work section.
>
> We would like to kindly clarify that we do **NOT** claim privacy auditing on worst-case data to be our main contribution. Our aim is to show that even with existing auditing methods, it becomes evident that there are numerous misleading claims regarding the use of synthetic data in training models.
>
> Breaking the four defenses we analyzed was not particularly challenging. This highlights that before researchers claim their empirical defenses to be privacy-preserving, they should take one additional step - conducting rigorous evaluations, as demonstrated in our work. Such evaluations are not difficult to perform. We hope this inspires future researchers to approach this issue more carefully, which is our primary goal.
>
> We hope that this work can bring these important concerns to the forefront of the community's attention and foster further discussions on avoiding misleading results in privacy.
>
> > Q2: Vulnerability of synthetic data against membership inference attacks has been considered in the literature.
>
> Thanks for the questions. We have reviewed the suggested work [1]. It primarily focuses on applying MIA to generative models, with the attack target being to determine whether a specific sample was used to train a GAN. This is quite different from our approach, where we use synthetic data generated from the original private dataset to train classification models. Our goal is to determine whether a classification model trained solely on synthetic data can protect privacy.
>
> > Q3: Results on more datasets could be provided, e.g., datasets from other domains.
>
> Thank you for suggesting reporting results on more datasets. While we agree that using more datasets would nevertheless have been interesting, we decided against this:
>
> (1) To do fair comparisons, we needed all studied methods to achieve reasonable test performance. Some methods, such as Dataset Distillation, perform poorly on more complex datasets like CIFAR-100 or ImageNet, which would make comparisons less meaningful.
>
> (2) Similar datasets (e.g., CIFAR-100) Figure 1 alone requires over 5800 GPU hours (without hyperparameters tuning and additional experiments). We decided that the auxiliary insights were not worth the cost.
>
> (3) Our approach aligns with previous studies like [a], which also conducted comprehensive evaluations of defense methods primarily on CIFAR-10. This consistency facilitates a direct comparison between our results and existing literature.
>
>
> [a] Evaluations of Machine Learning Privacy Defenses are Misleading. CCS 2024.
>
>
>
> > Q4: Do you think similar comparisons could be easily carried out in different image datasets or in other domains (e.g., tabular data) ?
>
> That's a good point! It's worth mentioning TabDDPM[b] as an example. In this approach, the researchers first train a diffusion model using private training data, then generate synthetic tabular data, and finally train a high-performing classifier using only the synthetic data. While the authors claim this method protects privacy, it hasn't been evaluated under rigorous privacy testing protocols.
>
> I believe this approach requires a more thorough privacy evaluation to validate these claims. We could also consider adding this experiment to the appendix based on your suggestions!
>
> [b] TabDDPM: Modelling Tabular Data with Diffusion Models. ICML 2023.

---

> > ### Author Response · Authors · 2024-11-18
> > **Reply to Reviewer qB2Z [2 / 2]**
> >
> > > Q5: Details of experiments in Figure 3 and Figure 4?
> >
> > Thank you for bringing this issue to our attention! We will include these details in the appendix to ensure clarity and address this point thoroughly.
> >
> > For Figure 3, the experiments were conducted on CIFAR-10, where we used 500 mislabeled samples to simulate the most vulnerable data. The attack settings follow those of LiRA. For each defense, we trained 16 models, ensuring that each sample only appears in the training set of half of the models. We used leave-one-out cross-validation—each time using one model as the victim and the remaining 15 models as the attacker's shadow models. The shadow models were implemented using a ConvNet architecture. We evaluated the privacy protection levels for both average-case and worst-case.
> >
> > Figure 4 is of the same experimental setting as the undefended baseline without any defense, but here we set the number of shadow models to 256 instead of 32. For the “average-case” subfigure, we select a normal sample and plot its loss distributions respectively when it is a member and when it is not a member. For the “most vulnerable” subfigure, we select a mislabeled canary and do the same thing.
> >
> > > Q6: When focussing on practical scenarios: do you think the situation would differ, if instead of using synthetic samples (e.g., mislabeled samples) for the auditing, you would try to find the most vulnerable data samples in the dataset?
> >
> > This is a great question!
> >
> > In a practical setting (specifically with natural data rather than strong canaries), the challenge lies in identifying the most vulnerable data in a given dataset without the need to train thousands of models. This is somewhat similar to [c]; definitely needs further investigation!
> >
> >
> > Indeed, we have ongoing work showing a very efficient method to identify the most vulnerable data. In this way, the situation would not differ much. While we cannot disclose further details about that work due to the double-blind review process, we can certainly add some discussions about this in the appendix at a later stage.
> >
> > [c] Privacy Auditing with One (1) Training Run. NeurIPS 2023.

---

> > > ### Comment · Reviewer_qB2Z · 2024-11-25
> > >
> > > Thank you for the replies! I understand these compute cost considerations are valid points for not carrrying out experiments on large image datasets. However, unfortunately I also think that conclusions relaying on empirical results found using one dataset are not yet that definite, some other smaller datasets (perhaps from other domains) would strengthen the paper, as also pointed out by reviewer ybiF.
> > >
> > > In addition to the paper by Hayes et al., 2019, there are also other papers on membership inference attacks on synthetic data, and speficially targeted on data and not the generative models, see, e.g., [Van Breugel et al., 2023](https://arxiv.org/pdf/2302.12580). It would bo good to mention some of these approaches.
> > >
> > > I think the paper is well written and nicely illustrate the need for DP in when fine-tuning generative models, however due to aforementioned reasons, I am keeping my score 5.

---

> > > > ### Author Response · Authors · 2024-11-25
> > > > **reply**
> > > >
> > > > Thank you for your reply!
> > > > Since privacy is not an average metric, demonstrating even a single instance where all these heuristic defenses fail to preserve privacy on this dataset is sufficient to highlight the need for more rigorous evaluations.

---

> > > > > ### Comment · Reviewer_qB2Z · 2024-12-02
> > > > >
> > > > > Thanks for the reply. This is a good point actually, I agree with you. In this sense the paper does its job. Although more example and datasets would strengthen the paper. After reconsideration, I have decided to raise my score by 1.
> > > > >
> > > > > Minor: there is a dot missing on line 778.

---

### Meta-Review · Area_Chair_yDs2 · 2024-12-20

**Metareview:**

The paper evaluates the empirical privacy protection provided by a number of synthetic data approaches without formal privacy in deep learning using membership inference attacks.

**Strengths:**
* Clearly written paper
* Illustrates an important point in relative strength of empirical privacy protection from different approaches

**Weaknesses:**
* Limited breadth of empirical evaluation
* Some reviewers considered the contribution to be limited

In extensive discussion between the reviewers, a consensus was reached to recommend acceptance.
The paper provides compelling concrete evidence on the vulnerability of heuristic privacy protection from synthetic data approaches.
The limited empirical evaluation was considered acceptable given the computational demands of the methods.

**Additional Comments On Reviewer Discussion:**

This was a borderline paper that was discussed at an online meeting between the AC and most reviewers.
The meeting reached a consensus recommending acceptance.

---

### Decision · Program_Chairs · 2025-01-22

Accept (Poster)